# Differences in Corticosterone Release Rates of Larval Spring Salamanders (*Gyrinophilus porphyriticus*) in Response to Native Fish Presence

**DOI:** 10.3390/biology11040484

**Published:** 2022-03-22

**Authors:** Amanda R. Bryant, Caitlin R. Gabor, Leah K. Swartz, Ryan Wagner, Madaline M. Cochrane, Winsor H. Lowe

**Affiliations:** 1Department of Biology, Texas State University, San Marcos, TX 78666, USA; arb326@txstate.edu; 2Montana Freshwater Partners, Livingston, MT 59047, USA; leahswartz9@gmail.com; 3School of Environment and Natural Resources, The Ohio State University Columbus, Columbus, OH 43210, USA; wagner.1286@buckeyemail.osu.edu; 4Division of Biological Sciences, University of Montana, Missoula, MT 59812, USA; madaline.cochrane@umontana.edu (M.M.C.); winsor.lowe@mso.umt.edu (W.H.L.)

**Keywords:** amphibian, stress physiology, coping capacity, glucocorticoids, predation

## Abstract

**Simple Summary:**

In amphibians, glucocorticoid hormones play a key role in the response to predation stress. Predators can directly affect prey via injury and death, but they can also have indirect effects due to the activity of glucocorticoids. The regulation of glucocorticoids can differ between populations that have co-evolved with predators and those that have not. We measured glucocorticoids at baseline and in response to a novel stressor in free-living larval salamanders that either live with or without fish predators naturally. We found that salamanders living with fish predators had lower measures of glucocorticoids than those without fish predators. Our study indicates that predator presence alters glucocorticoid regulation, which may allow species to better cope with native and introduced predators.

**Abstract:**

Invasive fish predators are an important factor causing amphibian declines and may have direct and indirect effects on amphibian survival. For example, early non-lethal exposure to these stressors may reduce survival in later life stages, especially in biphasic species. In amphibians, the glucocorticoid hormone corticosterone is released by the hypothalamo–pituitary–interrenal axis (HPI), as an adaptive physiological response to environmental stressors. The corticosterone response (baseline and response to acute stressors) is highly flexible and context dependent, and this variation can allow individuals to alter their phenotype and behavior with environmental changes, ultimately increasing survival. We sampled larvae of the spring salamander (*Gyrinophilus porphyriticus*) from two streams that each contained predatory brook trout (*Slavelinus fontinalis*) in the lower reaches and no predatory brook trout in the upper reaches. We measured baseline and stress-induced corticosterone release rates of larvae from the lower and upper reaches using a non-invasive water-borne hormone assay. We hypothesized that corticosterone release rates would differ between larvae from fish-present reaches and larvae from fish-free reaches. We found that baseline and stressor-induced corticosterone release rates were downregulated in larvae from reaches with fish predators. These results indicate that individuals from reaches with predatory trout are responding to fish predators by downregulating corticosterone while maintaining an active HPI axis. This may allow larvae more time to grow before metamorphosing, while also allowing them to physiologically respond to novel stressors. However, prolonged downregulation of corticosterone release rates can impact growth in post-metamorphic individuals.

## 1. Introduction

Amphibians are the most imperiled vertebrate group on Earth, with an estimated 43% of species currently in decline [1]. In the United States, amphibian populations are estimated to be decreasing at a rate of 3.7% annually [2]. As populations of vulnerable amphibians continue to decline, it is crucial that we identify the causes and mechanisms influencing these declines. Invasive fish predators are one of the major factors influencing amphibian declines, particularly at breeding sites [3,4]. Fish predators reduce amphibian populations directly via predation on larvae and adults [5,6]. They also have indirect effects on amphibian populations due to the costs of mounting a physiological response to predation, including altered larval behavior and morphology, reduced growth, impaired reproduction, and reduced recruitment [7,8,9,10,11,12,13]. These indirect effects can persist even after the immediate threat of predation has ended. Reduced growth and smaller size at metamorphosis can delay sexual maturity, reducing recruitment and leading to population level declines [14,15,16]. Some amphibian populations may lack the co-evolutionary history to mount an efficient response to non-native predator pressure [17,18,19], and little is known about how amphibians respond to fish predators even when they do have a shared co-evolutionary history [13]. Studying how amphibians respond to known fish predators could allow us to make future predictions about which amphibian populations will be more susceptible to invasive fish predators, and which populations will be physiologically able to cope with novel threats.

In amphibians, the hypothalamo–pituitary–interrenal (HPI) axis is the primary endocrine system controlling the physiological and behavioral response to external stressors via the regulation of corticosterone (CORT), the main glucocorticoid hormone in amphibians [20]. When individuals are exposed to acute or short-term stressors, circulating CORT levels frequently become elevated [21]. This adaptive temporary increase in CORT alters individual performance in a variety of ways, including changes in energy allocation via the mobilization of energy stores, decreased foraging, and reduced reproductive behaviors [22,23,24,25]. When exposure to stressors is chronic, individuals experience costs of repeatedly mounting a CORT response, and may eventually lose the ability to physiologically respond to further challenges [26].

Frequent exposure to stressors during the amphibian larval stage can affect individual performance and behavior in several ways. Chronically elevated CORT is associated with decreased size at metamorphosis, likely due to changes in metabolism and lipid storage [27,28,29]. Reduction in metamorphic size and lipid levels can lead to slower growth and lower survival rates during the terrestrial stage [30,31]. Additionally, frequent exposure to early life stressors can alter the CORT response to stressors at later life stages. Tadpoles exposed to frequent stressors grow more slowly and have downregulated CORT responses to handling stress, as well as reduced fat storage and growth rates as adults, although some individuals have exhibited catch-up growth depending on the stressor they experienced [32,33]. To maintain lifetime fitness, it is critical to mitigate the long-term costs of elevated CORT while retaining the ability to respond to new challenges.

The amphibian CORT response to predation is highly variable and context dependent [34,35]. Middlemis-Maher et al. [36] found that *Rana sylvatica* tadpoles had reduced whole-body CORT levels in response to acute exposure to chemical predator cues of a dragonfly larvae compared to the control treatment. This downregulation of the HPI axis in response to the threat of predation can lead to decreased movement and increased hiding behaviors [37]. *R. sylvatica* tadpoles exposed to dragonfly predatory cues throughout development also had lower stressor-induced whole body CORT levels than predator-naïve tadpoles, indicating an ability to downregulate the CORT response to cope with frequent stressor exposure [38]. Additionally, Davis and Gabor [19] found that *Eurycea nana* exhibited lower baseline and stressor-induced water-borne CORT release rates in response to frequently encountered fish predators when compared to individuals exposed to rarely encountered fish predators, indicating an ability for aquatic salamanders to differentially regulate the CORT response to known versus novel stressors. This flexibility of the CORT response allows individuals to alter their phenotype with the environment, increasing the probability of survival.

We studied the physiological response of spring salamanders (*Gyrinophilus porphyriticus*) to naturally occurring fish predators by comparing the stress response of larval salamanders in stream reaches with and without predatory fish. *G. porphyriticus* are members of the family Plethodontidae, the lungless salamanders. Larvae are exclusively aquatic, and the larval period can last up to seven years (M.M. Cochrane, unpublished data). Adults metamorphose and are still mainly aquatic, but can forage terrestrially [39]. This species inhabits small, cool, well-oxygenated streams along the Appalachian uplift. Between 1999 and 2018, adult abundance in one New Hampshire stream declined by roughly 50%; however, adult survival did not decrease and there was no change in larval abundance over that same time period, indicating a lack of successful adult recruitment and highlighting the metamorphic period as a critical life stage for the species [40]. Brook trout (*Salvelinus fontinalis*) are key native predators of larval salamanders throughout their range, and have been associated with decreases in density, size, and activity in several species [41,42,43,44]. At the Hubbard Brook Experimental Forest, in central New Hampshire, Lowe et al. (2018) found that in stream reaches where *S. fontinalis* are present, larval survivorship of *G. porphyriticus* is reduced; however, individual body condition is higher compared to individuals from reaches without *S. fontinalis* [45].

We collected water-borne baseline and stressor-induced hormone samples from individuals in four stream reaches at Hubbard Brook: two upstream reaches where *S. fontinalis* are absent (non-predator) and two downstream reaches where *S. fontinalis* are present (predator). The absence of fish in upstream reaches is likely due to physical barriers that prevent upstream movement, such as cascades and waterfalls. Because *S. fontinalis* are known predators of *G. porphyriticus*, we hypothesized that CORT release rates would vary as a function of chronic predator presence. If chronic predator presence is perceived as a chronic stressor, then we predicted higher baseline CORT release rates and no change from baseline CORT to agitation CORT release rates due to an inability to mount a response to an acute stressor. If individuals from predator reaches have physiologically adapted to frequent stressors, we predicted that salamanders would upregulate or downregulate their baseline CORT release rates compared to individuals from non-predator reaches, and agitation CORT release rates would be significantly higher than baseline CORT release rates in individuals from predator reaches.

## 2. Materials and Methods

Salamanders (*n* = 73) were collected from two first-order streams (Bear and Zigzag) in the Hubbard Brook Experimental Forest over a four-day period in August 2019 to minimize the possibility of weather biasing our results. We conducted these surveys in conjunction with standard mark-recapture surveys. We flipped 1 rock every meter for 500 m in each reach. After capture, individual larval salamanders were placed in 200 mL of spring water for 1 h to collect baseline CORT release rates. After 1 h, we transferred the individuals to a new container with 200 mL of water and agitated for 1 min every 3 min for another hour to collect stressor-induced CORT release rates. After stressor-induced CORT release rates were collected, we weighed and measured snout–vent length (SVL) for all individuals, and PIT tagged all unmarked individuals before releasing them at their capture locations.

Non-invasive water-borne hormone measures provide an integrated measure of corticosterone that is better for measuring chronic stress than point measures, such as from blood plasma [46]. Several studies have validated the use of water-borne hormones in multiple species of frogs and salamanders [47,48,49,50]. Non-invasive hormone monitoring provides tools to understand the ecology of amphibians in relation to environmental stressors with reliable and rapid assessments in the field [51].

All water samples were placed in coolers until they were returned to the laboratory. We stored hormone samples at −20 °C then thawed the samples at 4 °C prior to extraction. Hormone extractions were carried out using solid phase extraction columns (SepPak Vac3 cc/500 mg; Waters, Inc., Milford, MA, USA) following [48]. The SPE columns were primed with 4 mL of distilled water, followed by 4 mL of MeOH. The water samples were then passed completely through the SPE columns to extract the CORT from the water. After sample extraction, we eluted the extraction columns using 4 mL of 100% HPLC-grade methanol. We placed the eluted samples in a 37 °C water bath and used and EVAP-O-RAC to release a gentle stream of nitrogen over the liquid to evaporate the methanol from the samples, leaving behind a dry hormone residue. We resuspended the hormone residue in a mixture of 5% EtOH and 95% enzyme-immunoassay (EIA) buffer (Cayman Chemicals Inc., Ann Arbor, MI, USA) to achieve a total volume of 160 µL of solution. We vortexed the samples for 2 h to ensure thorough mixing, then plated all samples in duplicate using a Corticosterone EIA kits (No. 501320, Cayman Chemical Company, Inc., Ann Arbor, MI, USA; assay has a range of 8.2–5000 pg/mL and a sensitivity (80% B/B0) of approximately 30 pg/mL). Corticosterone values were measured using a spectrophotometer plate reader set at 405 nm (BioTek 800XS). Inter-plate variation was 13.2% (3 plates; intra-plate variation: 0.06–6.3%). We measured release rates over an hour by multiplying CORT values (pg/mL) from the plate analysis by the final resuspension volume (0.2 mL) then divided by mass (g) of the individual to control for differences in development due to size. We then natural log transformed these data to meet the assumption of normality for our model.

We performed our analysis using a linear mixed-effects model in R version 4.0.5 using the lme4 and lmerTest packages. In our initial model, stream, CORT type (baseline or agitation), and treatment (predator or non-predator) were fixed factors. However, the effect of stream was not significant, so was removed from the final model. Individuals were coded as a random factor nested within stream. We also constructed a simple linear model comparing SVL across treatments. Since there was no difference in SVL by treatment and we had already accounted for differences in development we did not include it as a factor in our analysis (Table 1). Hydrologic and water chemistry variables, including water temperature, food availability, and dissolved oxygen, do not vary at the scale of our study (i.e., between downstream and upstream reaches), so were not included in our analysis [52]. We plotted residuals and found no pattern in their distribution, so we concluded that our data met the assumption of homoscedasticity. We used a likelihood ratio test to compare our model against a reduced model containing only individuals as a random factor. We also performed a least-squares means comparison using the R package lsmeans to test linear contrasts among our fixed factors.

## 3. Results

Our full model explained significantly more variation in the data than the reduced model (χ^2^ = 11.245, *p* < 0.001). There was a significant effect of CORT type on CORT release rates, with larval salamanders showing a significant increase in stressor-induced CORT release rates over baseline (Intercept estimate = 4.09, Baseline CORT estimate = −0.46, *p* < 0.0001; Table 2). There was a significant effect of treatment, with individuals from predator reaches having significantly lower baseline and agitation CORT release rates than individuals from non-predator reaches (Intercept estimate = 4.09, Predator Treatment = −0.61, *p* = 0.0009; Table 2; Figure 1).

These results were consistent with results of our least-squares means comparison (Figure 1). There was a significant difference between baseline and agitation CORT release rates in the no predator treatment (*p* < 0.0001) and in the predator treatment (*p* < 0.0001). Baseline CORT release rates also significantly differed between the predator and no predator treatments (*p* = 0.0045). There was also a significant difference in agitation CORT release rates between the predator and no predator treatments (*p* = 0.0045).

## 4. Discussion

Fish are the most commonly introduced predators causing declines in amphibians at breeding sites, and stress from the presence of fish predators may have wide-ranging effects on amphibian populations [3,53,54]. These negative effects are not exclusive to introduced predators and understanding how amphibians cope with predation pressure from known threats can serve as a point of comparison for future studies on novel threats. We found that larval *G. porphyriticus* from reaches with predatory brook trout had lower baseline CORT release rates compared to individuals from fish-free reaches. Nonetheless, larvae from reaches with brook trout can still mount a stress response to acute stressors, indicating that they have maintained a responsive HPI axis.

Our study shows that *G. porphyriticus* larvae have altered CORT release rates and this may aid in coping with environmental differences in predator pressure. Individuals may cope with predation stress by downregulating CORT release rates; however, these alterations may have long-term effects during and after metamorphosis [55]. Because consistently elevated CORT can have lasting negative effects on size [56], body condition [57], and timing of metamorphosis [58], the observed downregulation of CORT in *G. porphyriticus* may adaptively allow individuals to cope with frequent stressors while avoiding the negative consequences of chronically elevated CORT. An additional factor may play a role in the observed changes in CORT. Specifically, the upper reaches tend to have higher abundances of *G. porphyriticus* than lower reaches [59]. Adult *G. porphyriticus* are likely the main alternative aquatic predator because they also eat larval salamanders of their own species, but at a lower frequency than *S. fontinalis* (W.L. personal obs.). Therefore, we could alternatively interpret that CORT is higher in the fish free reaches because of predation by conspecifics, or overall higher densities of salamanders. In another study, aquatic larval salamanders maintained at higher densities had higher CORT release rates [60], but the stress response was not measured, so it is not clear if these release rates were adaptive. Here, we found that larvae from fish free locations had both higher CORT at baseline and higher stress response compared to larvae from fish sites. Fish are more voracious predators than conspecifics, so we think this alternative hypothesis is less likely, but we would need further testing to differentiate these two hypotheses.

The observed differences in baseline and agitation CORT release rates of *G. porphyriticus* larvae indicates flexibility in response to stressors within the environment but may have carryover effects during and after metamorphosis, leading to variation in fitness and survival [58]. We did not observe any significant differences in size between treatments; however, the presence of *S. fontinalis* reduces growth in *G. porphyriticus* larvae [46,59]. Growth and development during the larval stage can have consequences for resource allocation and the response to stressors in post-metamorphic frogs, and variation in the larval environment can impact growth, feeding behavior, and CORT release rates in adult salamanders [33,60]. Larval salamanders maintained at higher densities had higher CORT release rates and that was correlated with smaller mass at metamorphosis than those with lower CORT release rates [60]. Downregulating CORT in the sites with fish predators may help offset these costs in terms of growth. However, downregulating CORT could have long-term costs that may be partially responsible for the lack of metamorphic recruitment in *G. porphyriticus* [41]. It may be that the ability to flexibly respond to predation pressure in the two different portions of the reaches may aid in coping with alternative stressors. Similarly, tadpoles from urban and agricultural ponds show different patterns of baseline and stress response that aid in coping with those different environments and the observed glucocorticoid flexibility is a plastic trait. Future research on how the CORT response of larvae influences growth, metamorphosis, and response to stressors at later life stages could provide valuable insight into the causes of amphibian declines.

While our findings do not provide direct insight on how amphibians respond to invasive fish predators, we see a similar pattern of response in two species of aquatic salamanders that have coevolved with fish. As with *G. porphyriticus*, *E. nana* also downregulated CORT in response to native fish predators. Furthermore, *E. nana* exhibited similar behavioral anti-predator responses to both novel and native fish predators, indicating an ability to mount a generalized antipredator response [18]. Overall, these results suggest that salamanders, as with tadpoles, show glucocorticoid flexibility which may aid in responding either native or introduced fish predators, but further work is needed to determine the taxonomic breadth of this ability. Additionally, with the potential to combine hormonal data collection with long-term mark-recapture efforts in the field, this study can serve as a starting point for future research on the long-term effects of exposure to fish predators on the CORT response in amphibians. This, along with studies that explore the relative cost of the presence of fish predators versus conspecific predators, could aid in more accurately measuring the indirect costs of fish predators.

## 5. Conclusions

We found that *G. porphyriticus* show flexibility in their glucocorticoid response to different predation pressures, similar to other species of salamanders and tadpoles. While the ability to alter their physiological response may minimize the costs of fish predators, we do not know the long-term costs or benefits of alternating their physiological response. Nonetheless, it is clear that the direct effects of fish predators are greater than the indirect effects which accounts for the observed declines in amphibian populations where fish are introduced. For species that do persist in the face of introduced fish predators, it is possible that amphibians will be able to flexibly alter their glucocorticoid response and thereby minimize the indirect effects of fish predators.

## Figures and Tables

**Figure 1 biology-11-00484-f001:**
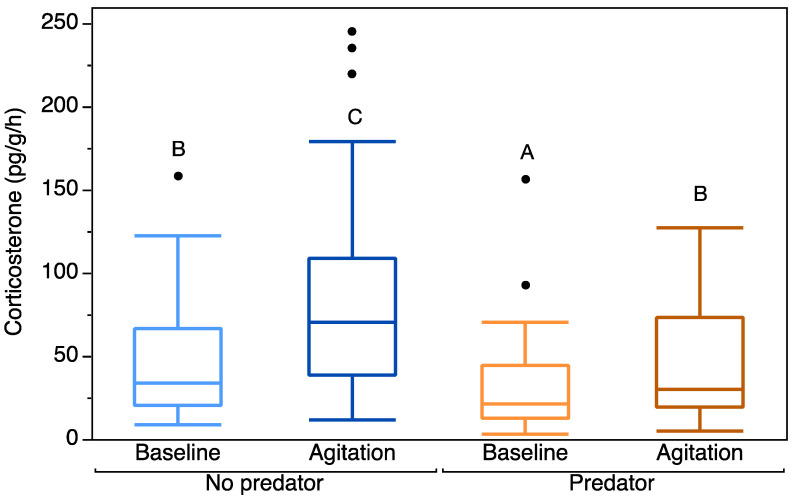
Mean baseline and agitation corticosterone release rates (pg/g/h) (untransformed) for individuals from stream reaches without fish predators and with fish predators. Box plots indicate median, range, and first and third quartiles. Points indicate outliers. Letters indicate significant differences in CORT release rates between combinations of fixed factors.

**Table 1 biology-11-00484-t001:** Sample size, mean, and standard error of snout–vent length (mm) and mass (mg) for each population. Treatment refers to reaches in each stream with and without fish predators.

*N*	Stream	Treatment	Mean SVL (mm)	SE	Mean Mass (mg)	SE
20	Bear	No predator	47.80	2.03	2.37	0.30
18	Zigzag	No predator	54.72	2.81	4.03	0.56
20	Bear	Predator	47.70	2.71	2.78	0.41
15	Zigzag	Predator	50.67	2.43	2.75	0.32

**Table 2 biology-11-00484-t002:** Mixed effects model output. Intercept reference categories are agitation CORT and no predator treatment. We found a significant effect of both CORT type (*p* < 0.0001) and treatment (*p* = 0.0009).

Fixed Effects	Estimate	SE	2.5% CI	97.5% CI	*p*-Value
(Intercept)	4.090	0.128	3.84	4.35	<0.0001
Baseline CORT	−0.468	0.081	−0.62	−0.31	<0.0001
Predator Treatment	−0.605	0.176	−0.96	−0.24	0.0009

## Data Availability

Data will be available from the Figshare Digital Repository.

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
