# Peer review of "Differences in Corticosterone Release Rates of Larval Spring Salamanders (Gyrinophilus porphyriticus) in Response to Native Fish Presence"

_biology, 2022, doi:10.3390/biology11040484_

Round 1

Reviewer 1 Report

Thank you for a thorough and thoughtful response to reviewer suggestions. I think the manuscript is greatly improved. 

Author Response

  • Reviewer 1. Thank you for a thorough and thoughtful response to reviewer suggestions. I think the manuscript is greatly improved.

    • Thank you. We appreciate your prior suggestions

Reviewer 2 Report

The authors present an innovative work on the association of physiological response of larval salamanders originating from risky (ie with fish) and safe habitat. They used a state of the art methodology of water-borne CORT assay they develop recently. The manuscript is there a good advance to our knowledge on the involved processes and of broad interest in biology. The manuscript reads well and is adequately finalized. I have therefore only minor suggestions.

Lines 54-59, as the ms on indirect non-consumptive effects of fish, it would be interesting to encompass the literature on the behavioural response of aquatic salamanders to introduced fish. Particualrly, the link between behavior and CORT is given in the second paragraph, so it would be good to introduce it in the first paragraph in the background of fish introduction.

Lines 59-60: which reference for this statement?

Lines 119, good to detail that adults are metamorphosed

Lines 133-136: somewhere in the manuscript, it would be interesting to state if there are stocking programs of these fish and if obstacles prevent colonisations in the reaches where the fish are absent

Line 155: redetail that you caught larvae only.

Table 1: Treatment seems to me misleading as this refers to population with and without fish and not salamanders artificially kept with and without predators. Just detailing it in the lend may be enough (as the authors did for figure 1).

Lines 255-260: what about the global effect of fish on the timing of metamorphosis of salamanders?

Line 280-283: larvae or paedomorph? Or just gilled if both

Author Response

Reviewer 2. The authors present an innovative work on the association of physiological response of larval salamanders originating from risky (ie with fish) and safe habitat. They used a state of the art methodology of water-borne CORT assay they develop recently. The manuscript is there a good advance to our knowledge on the involved processes and of broad interest in biology. The manuscript reads well and is adequately finalized. I have therefore only minor suggestions.

Lines 54-59, as the ms on indirect non-consumptive effects of fish, it would be interesting to encompass the literature on the behavioural response of aquatic salamanders to introduced fish. Particualrly, the link between behavior and CORT is given in the second paragraph, so it would be good to introduce it in the first paragraph in the background of fish introduction.

            -We added Behavioral response to this list and associated reference

Lines 59-60: which reference for this statement?

  • The references come with the detailed example in the next sentence.

Lines 119, good to detail that adults are metamorphosed

  • Added

Lines 133-136: somewhere in the manuscript, it would be interesting to state if there are stocking programs of these fish and if obstacles prevent colonisations in the reaches where the fish are absent

  • As noted on line 114, these are naturally occurring fish. They are not stocked. The presence or absence of fish is likely related to physical barriers along the stream channels (e.g., cascades and waterfalls that prevent upstream movement). We added this information here.

Line 155: redetail that you caught larvae only.

  • Good point. We changed this.

Table 1: Treatment seems to me misleading as this refers to population with and without fish and not salamanders artificially kept with and without predators. Just detailing it in the lend may be enough (as the authors did for figure 1).

  • We defined Treatment in the Methods, so we think it is valid to use terminology for simplicity. But we did edit the Table legend to clarify the use of this terminology.

Lines 255-260: what about the global effect of fish on the timing of metamorphosis of salamanders?

  • Good point. We changed “to cope” to “and this may aid in coping”

Line 280-283: larvae or paedomorph? Or just gilled if both

  • We changed the sentence to say two species of “aquatic” salamanders

Reviewer 3 Report

The authors of “Differences in corticosterone release rates of larval spring salamanders (Gyrinophilus porphyricticus) in response to native fish presence” offer a straightforward and interesting field-based study of the stress response of an amphibian to fish predator presence. That being said, the question being addressed, common methods that were applied, and single-species nature of the study are not likely of broad interest. Their methods were generally sound, though the use of lsmeans seems redundant given the existence of only two categories in each factor. My primary concerns are in regard to the Discussion section. The narrative feels incomplete given the framing of their work, and I believe that further interpretation of their results is needed. Line specific comments are as follows:

Lines 97-99: Reduced in comparison to what exactly?

Lines 141-143: A couple of typos here, I think.

Lines 150-152: What other predators were present in these streams? Were the distributions of these predators equivalent across streams, or could they have had a confounding effect on your results? Were there any other stressors evidently present at one stream reach but not the others?

Lines 155-156: Are these really “baseline” since the animal was just captured?

Lines 199-202: It seems like you should mention this before describing your final model structure.

Results: As a style issue, avoid using “we found” to start most sentences. Try diving in, like, “Our full model explained…”. Also, I’m not sure what the posthoc analysis results add (Lines 222-229). Since you only have two categories in each factor, the significance of factors reported in the main model output can only refer to differences between those two categories.

Figure 1: Is it true that there were no predators when trout were absent? This seems unlikely. I would specify that you are only talking about the fish species of interest here.

Lines 270-271: There are now several studies also showing the relationship between body size and juvenile survival in various salamander species, which I think underscore your point here.

Line 273: Metamorphic recruitment (i.e., survival to metamorphosis) and/or adult recruitment (i.e., survival of juveniles), it sounds like.

Lines 277-283: You have emphasized twice to this point how your data can inform possible responses of amphibians to fish invasion, yet you do not thoroughly interpret your results in this light. Do your results tell you that amphibian populations not previously exposed to fish will be at greater risk following fish introduction than experienced populations? Or could experienced amphibian populations be desensitized such that direct mortality is higher following introduction compared with naïve amphibian populations? Do you or previous others see a way to mitigate this threat in service of amphibian conservation? Given the emphasis put on this point in the text, clear interpretations and future directions are needed.

Lines 286-289: I find this statement vague. To address what question(s)? How exactly can these two methods be combined to address a gap in our knowledge.

Author Response

Reviewer 3. 

The authors of “Differences in corticosterone release rates of larval spring salamanders (Gyrinophilus porphyricticus) in response to native fish presence” offer a straightforward and interesting field-based study of the stress response of an amphibian to fish predator presence. That being said, the question being addressed, common methods that were applied, and single-species nature of the study are not likely of broad interest. Their methods were generally sound, though the use of lsmeans seems redundant given the existence of only two categories in each factor. My primary concerns are in regard to the Discussion section. The narrative feels incomplete given the framing of their work, and I believe that further interpretation of their results is needed. Line specific comments are as follows:

Lines 97-99: Reduced in comparison to what exactly?

  • We added “compared to the control treatment”

Lines 141-143: A couple of typos here, I think.

  • We decided to reword these last two sentences a bit to the following:

“If chronic predator presence is perceived as a chronic stressor, then we predicted higher baseline CORT release rates and no change from baseline CORT to agitation CORT release rates due to an inability to mount a response to an acute stressor. If individuals from predator reaches have physiologically adapted to frequent stressors, we predicted that salamanders would upregulate or downregulate their baseline CORT release rates compared to individuals from no-predator reaches, and agitation CORT release rates would be significantly higher than baseline CORT release rates in individuals from predator reaches.”

Lines 150-152: What other predators were present in these streams? Were the distributions of these predators equivalent across streams, or could they have had a confounding effect on your results? Were there any other stressors evidently present at one stream reach but not the others?

  • Spring salamanders eat larval salamanders of their own species, so they would be the other major predator in the water besides brook trout. The upper reaches tend to have higher abundances of salamanders than lower reaches, so perhaps there is more predation pressure from conspecifics on small larval salamanders at those sites. We address this in the Discussion. There are no known insect predators, but snakes may be predators and they are found in both downstream and upstream reaches.

  • This is what we added to the second paragraph of the discussion”

“An additional factor may play a role in the observed changes in CORT. Specifically, the upper reaches tend to have higher abundances of G. porphyricticus than lower reaches. Adult G. porphyricticus are likely the main alternative aquatic predator because they also eat larval salamanders of their own species, but at a lower frequency than S. fontinalis (W.L. personal obs.). Therefore, we could alternatively interpret that CORT is higher in the fish free reaches because of predation by conspecifics, or overall higher densities of salamanders. In another study, aquatic larval salamanders maintained at higher densities had higher CORT release rates [60], but the stress response was not measured, so it is not clear if these release rates were adaptive. Here, we found that larvae from fish free locations had both higher CORT at baseline and higher stress response compared to larvae from fish sites. Fish are more voracious predators than conspecifics, so we think this alternative hypothesis is less likely, but we would need further testing to differentiate these two hypotheses.”

Lines 155-156: Are these really “baseline” since the animal was just captured?

  • These are a relative baseline. All animals are in the same state of some level of stress from capture. We have explained this in earlier publications. Also, please note that water-borne hormones are an integrated measure of CORT that, unlike point measures from plasma, reflect some amount of prior experiences. However, we can also see an increase in CORT due to agitation over an hour time period because this is more of an acute stressor.

Lines 199-202: It seems like you should mention this before describing your final model structure.

  • While we see your point about moving this closer to the model, we decided to move it up to the end of the prior paragraph.

Results: As a style issue, avoid using “we found” to start most sentences. Try diving in, like, “Our full model explained…”. Also, I’m not sure what the posthoc analysis results add (Lines 222-229). Since you only have two categories in each factor, the significance of factors reported in the main model output can only refer to differences between those two categories.

  • Good point. We cleaned up the sentences and removed “we found” from our results section. As for the posthoc analyses, they indicate that the agitation of salamanders from predator reaches was not higher than baseline with no predators. We have now added a short interpretation about this comparison to the Discussion, based on some of your other comments.

Figure 1: Is it true that there were no predators when trout were absent? This seems unlikely. I would specify that you are only talking about the fish species of interest here.

  • We do specify fish predators in the figure legend. As per our methods, we define Treatment as with or without fish predators, so we use these words in our figure

Lines 270-271: There are now several studies also showing the relationship between body size and juvenile survival in various salamander species, which I think underscore your point here.

  • Thanks

Line 273: Metamorphic recruitment (i.e., survival to metamorphosis) and/or adult recruitment (i.e., survival of juveniles), it sounds like.

  • Metamorphic recruitment is correct

Lines 277-283: You have emphasized twice to this point how your data can inform possible responses of amphibians to fish invasion, yet you do not thoroughly interpret your results in this light. Do your results tell you that amphibian populations not previously exposed to fish will be at greater risk following fish introduction than experienced populations? Or could experienced amphibian populations be desensitized such that direct mortality is higher following introduction compared with naïve amphibian populations? Do you or previous others see a way to mitigate this threat in service of amphibian conservation? Given the emphasis put on this point in the text, clear interpretations and future directions are needed.

  • While we cannot address behavioral response to fish invasion with this paper, we can address physiological costs. We have now added the following Conclusion section that links the points about mortality and responsiveness to fish. I don’t think we can suggest ways to mitigate threats given our experiment. Instead, we can indicate that species may be able to show flexible glucocorticoid response that could decrease the indirect effects of fish predators.

“We found that G. porphyricticus show flexibility in their glucocorticoid response to different predation pressures, like other species of salamanders and tadpoles. While the ability to alter their physiological response may minimize the costs of fish predators, we do not know the long-term costs or benefits of alternating their physiological response. Nonetheless, it is clear that the direct effects of fish predators are greater than the indirect effects, which accounts for the observed declines in amphibian populations where fish are introduced. For species that do persist in the face of introduced fish predators, it is possible that amphibians will be able to flexibly alter their glucocorticoid response and thereby minimize the indirect effects of fish predators..”

Lines 286-289: I find this statement vague. To address what question(s)? How exactly can these two methods be combined to address a gap in our knowledge.

  • We added the following sentence in response to your comment, and to help clarify next

“This, along with studies that explore the relative cost of the presence of fish predators versus conspecific predators, could aid in more accurately measuring the indirect costs of fish predators.”